# Quantitative Proteomic and Phosphoproteomic Profiling of Lung Tissues from Pulmonary Arterial Hypertension Rat Model

**DOI:** 10.3390/ijms24119629

**Published:** 2023-06-01

**Authors:** Ang Luo, Yangfan Jia, Rongrong Hao, Yafang Yu, Xia Zhou, Chenxin Gu, Meijuan Ren, Haiyang Tang

**Affiliations:** 1College of Veterinary Medicine, Northwest A&F University, Xianyang 712100, China; jiayangfan0425@163.com (Y.J.); hr172052@163.com (R.H.); yuyafang2020@163.com (Y.Y.); zhouxia1336508@163.com (X.Z.); gu.chenxin00@gmail.com (C.G.); 2Life Science Research Core Service, Northwest A&F University, Xianyang 712100, China; renmeijuan@nwafu.edu.cn

**Keywords:** pulmonary arterial hypertension, proteomic, phosphoproteomic

## Abstract

Pulmonary arterial hypertension (PAH) is a rare but fatal disease characterized by elevated pulmonary vascular resistance and increased pressure in the distal pulmonary arteries. Systematic analysis of the proteins and pathways involved in the progression of PAH is crucial for understanding the underlying molecular mechanism. In this study, we performed tandem mass tags (TMT)-based relative quantitative proteomic profiling of lung tissues from rats treated with monocrotaline (MCT) for 1, 2, 3 and 4 weeks. A total of 6759 proteins were quantified, among which 2660 proteins exhibited significant changes (*p*-value < 0.05, fold change < 0.83 or >1.2). Notably, these changes included several known PAH-related proteins, such as Retnla (resistin-like alpha) and arginase-1. Furthermore, the expression of potential PAH-related proteins, including Aurora kinase B and Cyclin-A2, was verified via Western blot analysis. In addition, we performed quantitative phosphoproteomic analysis on the lungs from MCT-induced PAH rats and identified 1412 upregulated phosphopeptides and 390 downregulated phosphopeptides. Pathway enrichment analysis revealed significant involvement of pathways such as complement and coagulation cascades and the signaling pathway of vascular smooth muscle contraction. Overall, this comprehensive analysis of proteins and phosphoproteins involved in the development and progression of PAH in lung tissues provides valuable insights for the development of potential diagnostic and treatment targets for PAH.

## 1. Introduction

Pulmonary arterial hypertension (PAH) is a progressive disease defined by a standard in which the mean pulmonary artery pressure (mPAP) exceeds 20 mmHg at rest, according to the 6th World Symposium on Pulmonary Hypertension [1]. The progression of PAH is marked by increasing pulmonary vasoconstriction and vascular remodeling, leading to the narrowing of vessel lumen and heart hypertrophy and, eventually, resulting in right heart failure, the leading cause of mortality in PAH patients [2]. Pathologically, the development of PAH involves a complex interplay between pulmonary arterial smooth muscle cells (PASMC), endothelial cells (PAEC), fibroblasts and inflammatory cells [3]. Etiologically, PAH can be idiopathic or attributed to factors such as genetic mutation (e.g., *BMPR2*, *TBX4* and *Sox17*)*,* drugs and toxins, connective tissue disease and congenital heart disease, or even HIV infection [4,5]. Although numerous proteins or signaling pathways have been implicated in the development and progression of PAH [6], such as STAT3 (signal transducer and activator of transcription 3) [7], hypoxia-inducible factors (HIFs) [8] and interleukin-6 (IL6) [9], studies focusing on individual protein or pathway have limitations due to the high complexity of PAH and potential bias.

With the advances in instrumentation and the development of bioinformatics, mass spectrometry-based proteomic studies have emerged in the field of PAH and are garnering increasing attention [10]. Novel proteomic strategies enable the global analysis of groups of proteins associated with the development and progression of PAH. For example, using label-free quantification proteomics, Vahitha B. Abdul-Salam and colleagues identified 25 differentially expressed proteins in lung tissues from PAH patients [11]. In addition, there are also proteomics studies on the isolated pulmonary arteries (PAs) from hypoxia-induced PAH rat models or human PAECs from patients with PAH, to identify potential protein markers of PAH [12,13]. However, these studies have inherent limitations. Most of the clinical samples and the rat PAs were collected at the end-stage of the diseases, leading to the identified proteins tending to be passenger rather the driver proteins in PAH.

To dynamically investigate the proteins and signaling pathways involved in different stages of PAH, this study used monocrotaline (MCT)-induced PAH rat model and performed TMT (tandem mass tag)-based quantitative proteomic analysis. The lung proteome of rats injected with MCT from one to four weeks were analyzed. Phosphorylation, as a crucial posttranslational modification, plays major roles in regulating protein activity and functions. Many phosphorylated proteins, including ACE2 (angiotensin-converting enzyme 2), AKT1, endothelial NOS (eNOS) and forkhead box O1 (FoxO1) [14,15,16,17], have been implicated in PAH development. To comprehensively study the phosphorylated proteins associated with PAH, quantitative phosphoproteomic profiling was also performed on the lung tissues from 3 weeks induced PAH rats. Furthermore, bioinformatic analyses were employed to identify the signaling pathways associated with PAH.

## 2. Results

### 2.1. Workflow for Total Proteome Analysis and Verification of PAH Animal Model

To dynamically elucidate the molecular mechanism underlying PAH, we used MCT-treated SD rats as the model and applied TMT-based quantitative proteomic profiling to study the protein changes in the lung tissues. Experimental design and workflow are summarized in Figure 1a. After a period of 1, 2, 3 and 4 weeks after MCT injection, rats were anesthetized for RVSP measurement; then, the lungs were isolated for MS sample preparation and histology analysis and the hearts were used for Fulton Index determination. At each time point, four control rats and four MCT rats were collected for analysis and sample preparation. The mean RVSP, Fulton index and RV/BW ratios for MCT rats keep increasing from 1 to 4 weeks after injection, while these parameters for the control rats stay relatively stable through the experiment (Figure 1b–d). In addition, compared to the control rats, HE staining of the lung tissues revealed that MCT treatment apparently increased the pulmonary arteries wall thickness (Figure 1e). All these hemodynamic changes indicated successful establishment of the PAH rat models. For mass spectrometry (MS) sample analysis, total proteins from lung tissue homogenates were quantified, reduced, alkylated and digested with Lys C and trypsin. Desalted peptides from each rat were labeled with one TMT channel, as indicated in Figure 1a. After TMT labeling, all the peptides from the same rat group were combined and desalted. The TMT-labeled peptides mixture was then subjected to fractionation and 12 final fractions were submitted to LC–MS/MS analysis; the MS raw data were analyzed with Proteome Discoverer software 2.2.

### 2.2. Proteomic Profiling of Lung Tissues from MCT Rat Model

Based on the experimental workflow and design shown in Figure 1a, we acquired four groups of proteomic data, which were named as Week 1, Week 2, Week 3 and Week 4. A total of 6901 proteins were identified and 6759 proteins were quantified; of these, 4263 quantified proteins were shared among 4 groups of data (Appendix A). Principal component analysis (PCA) of these data showed that MCT samples can be clearly separated from the control samples (Figure 2a and Appendix A). To find out the proteins whose expression significantly changed in the MCT rats compared to the control rats, a two-sample *t*-test was applied to the normalized protein abundances and volcano plots were used to display the results (Figure 2b). As shown in Figure 2b, even one-week MCT treatment caused the upregulation of 288 proteins and downregulation of 157 proteins in the MCT samples, though the hemodynamic terms (RVSP, Fulton index and RV/BW ratios) remained comparable to the control rats (Figure 1b–d and Figure 2b). Throughout the experiment, the consistent change in these hemodynamic terms indicated increasing severity of the symptoms, which was reflected by the increasing numbers of changed proteins from Week 1 to Week 3. In total, we identified 1333 significantly upregulated (*p*-value < 0.05, fold change (FC) > 1.2) proteins and 1327 downregulated proteins (*p*-value < 0.05, FC < 0.83) (Figure 2c,d). The FC of most proteins was less than two (−1 < log_2_(FC) < 1), while Week 3 and Week 4 had more highly changed proteins (|log_2_(FC)| >1) than Week 1 and Week 2 (Figure 2f). To obtain a general view of all these changed proteins, abundance of proteins significantly changed in at least one group of samples were shown in a heatmap after hierarchical clustering (Figure 2e). In addition, abundances of top-10 upregulated and downregulated proteins of each group were shown in detailed heatmaps (Appendix A). Overall, most of these top-10 changed proteins were group-specific, indicating quick dynamic changes for these proteins during the progression of PAH. Among these differently regulated proteins, we found some known PAH-associated proteins, such as resistin-like alpha (Retnla) [18], arginase-1 (Arg1) [19], Chymase 1 (Cma1) [20] and haptoglobin (HP) [21]; their relative abundances are shown in Figure 3a–d.

Recently, Xiao et al. published a study about RNA-sequencing analysis in lung tissues from rats treated with MCT for 1, 2, 3 and 4 weeks [22]. Venn diagrams were drawn to compare the significantly changed genes identified by Xiao et al. and the significantly changed proteins identified our study (Appendix A). Strangely, no overlap is observed between these two studies for rats treated with MCT for one week. From Week 2 to Week 4, the number of proteins shared by these two studies keeps increasing. Representative common upregulated genes or proteins include Arg1(arginase 1), chymase 1 (Cma1), fibronectin 1 (Fn1) and platelet factor 4 (Pf4); representative common downregulated genes or proteins include vasoactive intestinal peptide receptor 1 (Vipr1), angiotensin I converting enzyme (Ace I) and calcitonin receptor-like receptor (Calcr1) (Appendix A). In addition, proteomic analysis has also been performed in pulmonary arteries of hypoxia-induced PAH model rats by Zhang et al. [12]. We compared our data to the results of Zhang et al. and identified 21 common downregulated proteins, such as synemin (Synm), LIM zinc finger domain-containing 2 (Lims 2) and tropomyosin 1 (Tpm 1), and 14 common upregulated proteins, such as sideroflexin 1 (Sfxn1), Fn1 andthrombospondin 4 (Thbs4) (Appendix A).

### 2.3. Validation of Representative Proteins Revealed by Proteomic Analysis

To validate these significantly changed proteins identified above, Western blot (WB) was used to examine the expression of some representative proteins. One key purpose of this study is to reveal the proteins and pathways changed at the early stage of PAH development. KEGG pathway analysis revealed that terms such as cell cycle, DNA replication and DNA repair were specifically enriched for proteins upregulated in Week 1, such as Aurora kinase B (Aurkb), Cyclin-A2 (Ccna2) and several mini-chromosome maintenance proteins (Mcm2, 3, 4, 6, 7), which usually function as DNA helicase or replication licensing factor (Figure 3e). Interestingly, Aurkb and Ccna2 are two of the top-ten increased proteins in Week-1 (Appendix A). WB proved that the protein levels of Aurkb and Ccna2 were indeed significantly increased in lung tissues of rats treated with MCT for one week (Figure 3g,h). In addition, the expression of another top-10 increased protein Racgap1 (Rac GTPase Activating Protein 1) was also confirmed using WB (Figure 3g,h).

To dynamically show the proteins changed from one week to four weeks after MCT injection, protein clustering analysis was performed based on fold changes (FC) for proteins significantly changed in at least one week (Appendix A). Totally, all the proteins were arbitrarily divided into six clusters, of which we were especially interested in the proteins where FC value decreased (Cluster 4, 358 proteins) or increased (Cluster 1, 262 proteins) continuously. Pathway analysis revealed proteins with continuously increased FC were enriched in lysosome, phagosome and carbohydrate metabolic process, while proteins with continuously decreased FC were enriched in regulation of protein polymerization, negative regulation of catalytic activity and fluid shear and atheroscierosis. Next, expression of one potential PAH-associated protein called PLVAP, which belonged to Cluster 4, was verified by using WB in lung tissues of rats treated with MCT or PBS (Cont) for two or four weeks (Figure 3i,j).

### 2.4. Gene Ontology and Pathway Analysis of Significantly Changed Proteins

To understand which biological processes or pathways were affected during the progression of PAH, we performed GO and KEGG pathway analysis for proteins differently expressed between control and MCT samples. Proteins with FC > 1.2 or FC < 0.83 in each group were involved in the analysis. For upregulated proteins, adaptive immune response was the only biological process shared by all four groups of samples. Negative regulation of proteolysis and humoral immune response were two terms enriched in weeks 1, 3 and 4 (Appendix A). For downregulated proteins, no GO term was enriched in all four groups, while cell–cell junction organization and cellular modified amino acid metabolic process were two terms enriched in three groups of samples (Appendix A). KEGG pathway analysis of the upregulated proteins revealed that complement and coagulation cascades, ferroptosis, coronavirus disease (COVID-19) and lysosome were the terms enriched in at least three groups of data (Figure 4a). Notably, the involvement of complement and coagulation cascades in PAH has also been revealed in recent studies from other groups [12,23]. Moreover, there were also several pathway terms enriched in two groups, such as DNA replication, amino sugar and nucleotide sugar metabolism and glycerophospholipid metabolism (Figure 4a). Multiple pathways were enriched in at least two groups of data for downregulated proteins, such as apoptosis, endocytosis, bacterial invasion of epithelial cells, tight junction and Rap1 signaling pathway (Figure 4b).

In eukaryotic cells, compartmentalization makes various biological processes highly organized and functions of proteins are tightly associated with their cellular distribution. Protein–protein interaction is the basic way for proteins to take part in these biological processes. Next, to globally understand the functions of the significantly changed proteins identified here, we performed compartment analysis combined with protein–protein interaction network analysis. To make the analysis concise and confident, we only included proteins with |log_2_(FC)| > log_2_1.5 and *p*-value < 0.05 in at least two groups and proteins with continuously increased or decreased FC for at least three weeks. Surprisingly, most of the plasma membrane-localized proteins were downregulated, such as Tek receptor tyrosine kinase, plasmalemma vesicle-associated protein (PLVAP) and bone marrow stromal cell antigen 2 (Bst2) (Appendix A). However, ER, Golgi and endosome contained many more upregulated proteins than downregulated proteins. In addition, there were still large numbers of proteins that had no specific cellular localization and, interestingly, these upregulated proteins tended to interact with each other but not the downregulated proteins.

### 2.5. Phosphoproteomics Profiling of Lung Tissue from MCT Rat Model

Phosphorylation plays important roles in regulating protein structure and function and is often associated with human diseases. Next, we globally analyzed the phosphoproteins which were changed during the progression of PAH. Based on the proteomic analysis results, it seemed that there were more significantly changed proteins in the lung tissues of rats treated with MCT for 3 weeks than that in rats treated with MCT for 1, 2 or 4 weeks. Hence, we performed phosphoproteomics profiling of lung tissue from rats treated with MCT for 3 weeks. Experimental design and sample preparation procedures before phosphopeptides enrichment were described above (Figure 1). To enrich the phosphopeptides, the original 12 fractions of TMT-labeled peptides for proteome analysis were further combined into 6 final fractions. The peptide mixture was first enriched using TiO_2_ resin and next by Fe-NTA resin and, finally, all elutes were treated as different fractions for LC–MS/MS analysis (Figure 5a). Totally, 7436 peptides were identified, of which 6246 peptides were quantified and phosphorylated. PCA analysis of these phosphopeptides’ abundances showed that MCT samples could be clearly separated from the control samples, indicating significant differences between treatments (Figure 5b). Before statistical analysis, abundance of each phsophopeptide was normalized to the abundance of the corresponding protein. Student’s *t*-test identified 1412 upregulated phosphopeptides (630 phosphoproteins) and 390 downregulated phosphopeptides (271 phosphoproteins) in the MCT samples (Figure 5c,d). Venn diagrams showed more than half of these significantly changed phosphoproteins were not caused by changes in the corresponding unmodified proteins (Appendix A).

Next, comprehensive pathways enrichment analysis was performed for the corresponding proteins of these changed phosphopeptides (Figure 5e). Tight junction and adherens junction are two terms enriched by both upregulated phosphoproteins and downregulated phosphoproteins. Pathways specifically enriched by upregulated phosphoproteins include endocytosis, focal adhesion, ErbB signaling and vascular smooth muscle contraction, in which representative phosphorylated proteins included Myh 9/10/11/14 (myosin heavy chain), Pla2g4a (cytosolic phospholipase A2) and Araf (A-Raf proto-oncogene).

Finally, we integratively analyzed the proteomic data and phosophoproteomic data of the lung samples from rats treated with MCT for three weeks. PPI functional networks were created with significantly upregulated kinase proteins from proteomic data and upregulated phosphoproteins from phosophoproteomic data (Figure 6). In this network, phosphoproteins interacting with a certain kinase could be regarded as the potential substrates and their changed phosphorylation was likely to be caused by changes in the level of the corresponding kinase. In the upregulated group of the lung, epidermal growth factor receptor (Egfr), tyrosine–protein kinase (Btk) and serine/threonine–protein kinase (Plk2) were three representative kinases interacting with multiple phosphorylated proteins, such as mTor, Stat3, Tensin-1 (Tns1) and Stim2 (Figure 6). Of note, a recent paper also found increased Btk in lung tissues of MCT-induced PAH rats and inhibition of Btk attenuated MCT-induced PAH [24].

## 3. Discussion

The many similarities in pathogenesis between PAH and human cancers, such as altered crosstalk between cells in the vessels, inhibited cell death and sustained cell proliferation, have led to PAH being considered as a cancer-like disease [25]. Recently, empowered by the advances in instrumentation and data processing methods, large-scale deep proteomic profiling integrated with high-throughput sequencing has been used to comprehensively analyze the signaling pathways associated with carcinogenesis [26,27,28]. However, such comprehensive studies are still lacking in the field of PAH. In this study, using MCT-induced rat models, we performed temporal quantitative proteome profiling of the lung tissues at different stages of PAH. We quantified a total of 6759 proteins, of which 2660 proteins were significantly changed between control and MCT samples. To our knowledge, this is the most in-depth proteomic analysis conducted in PAH to date [11,12,29]. Among these changed proteins, 24 upregulated proteins and 37 downregulated proteins were consistently observed across all four groups of data, indicating their involvement from the early stages through the progression of PAH. GO and KEGG analyses revealed these 24 upregulated proteins were enriched in adaptive immune response and complement and coagulation cascades, highlighting the association between PAH and the immune system [30].

One advantage of this study is that we analyzed the proteome of rats treated with MCT for one week, at which time the hemodynamic parameters were comparable between control and MCT rats. Surprisingly, we identified 288 upregulated and 157 downregulated proteins. Notably, cell cycle and DNA replication proteins such as Aurkb and Ccna2 were increased specially in Week 1 MCT samples. The upregulation of these proteins likely contributes to the proliferation of smooth muscle cells and endothelial cells in pulmonary arteries. Furthermore, we observed changes in the expression of certain proteins, such as PLVAP, as the duration of MCT treatment increased. PLVAP, known as a component of the fenestral and stomatal diaphragms in fenestrated endothelia [31,32], is involved in maintaining the integrity of caveolae and controlling endothelial permeability. In addition to PLVAP, caveolin-1 and caveolin-2 are also components of the caveolae. In mice, loss of caveolin-1 resulted in vascular abnormalities and pulmonary hypertension, which were associated with damaged caveolae structure [33,34]. Endothelial cell-specific deletion of PLVAP in mice leads to abnormal caveolar and increased endothelial permeability [35]. Therefore, the potential roles of PLVAP in PAH may be associated with its function for maintaining the integrity of caveolar and controlling endothelial permeability. Here we analyzed the PAH-related proteins in the lung tissue of MCT-treated rats but we were not clear about the localization of these significantly changed protein in the lung. To establish the relationship between certain proteins with the development of PAH, further analysis of their expression in the pulmonary arteries using immunoblot or immunofluorescence is necessary.

In addition to proteomic analysis, we conducted phosphoproteomic profiling in the MCT rat model. In the lung tissues, we identified 630 upregulated phosphoproteins and 271 downregulated phosphoproteins. One of the major upregulated phosphoproteins was microtubule-associated protein 4 (Map4) and 9 phosphopeptides were identified. As a cytosolic protein, MAP4 is involved in microtube polymerization and its phosphorylation has been reported to affect cardiac microvascular density, endothelial cell migration and proliferation under hypoxia [36,37]. Endothelial cells’ dysfunction is an important character of PAH and increased MAP4 phosphorylation may play role in this process. In addition, one of the known PAH-related phosphoproteins Stat3 (Ser-727) was also found to be upregulated (−log_10_*p*-value = 5.89, FC = 1.5 or log_2_FC = 0.6) in our data (Appendix A), indicating consistency between our findings and previous publications. 

In terms of pathway analysis, while downregulated proteins were poorly enriched in pathways, the upregulated proteins were enriched in several important pathways implicated in PAH, including vascular smooth muscle contraction and ErbB signaling pathway [38]. The development of PAH is often characterized by increased proliferation of pulmonary arterial smooth muscle cells and dysregulated vascular smooth muscle contraction. The observed increase in proteins’ phosphorylation suggests a direct involvement of protein phosphorylation in the development of PAH. Supporting the enrichment of ErbB pathway, Egfr, the hub protein of this pathway, was found to be significantly upregulated in our proteomic profiling of the samples (Figure 3f). PPI functional network analysis revealed multiple upregulated phosphoproteins that interacted with Egfr, potentially serving as its substrates (Figure 6). Notably, our phosphoproteomic analysis also uncovered an intriguing finding: the upregulation of phosphorylated bone morphogenetic protein receptor type II (Bmpr2) at Ser 940 in the lungs from MCT rats. Bmpr2 is a highly important protein associated with both hereditary and non-hereditary PAH [39,40], yet its phosphorylation has not been reported previously. Exploring the potential effects of phosphorylated Bmpr2 function would be of great interest.

In summary, this study represents the first integration of proteomic and phosphoproteomic analysis of the lung tissues in PAH. However, there are certain limitations to consider. Some key PAH-related players, such as STAT3, HIF1α, IL6 and phosphor-STAT3, were not identified, likely due to their lower expression levels of proteins or phosphorylation compared to identified proteins in this study. Additionally, the depth of proteomic analysis could be enhanced by optimizing the method of fractionation. Another limitation of this study is lack of measurement of the cardiac output and total pulmonary resistance, which are crucial indicators for characterizing models alongside RVSP and Fulton index. Despite these limitations, our results contribute to a better understanding of the underlying molecular mechanism involved in the development and progression of PAH. Undoubtedly, the significant changes in these observed proteins and phosphoproteins should be validated in future clinical studies.

## 4. Materials and Methods

### 4.1. Rat Monocrotaline (MCT) Model 

Specific pathogen-free (SPF)-grade male Sprague-Dawley (SD) rats (6–7 weeks old) were purchased from Vital River Laboratories (Beijing, China) and housed under temperature-controlled (22–25 °C) and humidity-monitored conditions on a 12 h-dark and -light cycle. All rats were handled according to guidelines of the Institutional Animal Care and Use Committee of Northwest A&F University, and Guangzhou Medical University. After 48 h from their reception, the rats were randomly assigned into two group, control group and MCT group, and then weighed (the weight of all animals was between 200 g and 220 g). MCT group rats were intraperitoneally injected with 50 mg/kg dose of MCT (MedChemExpress, Shanghai, China) and control rats were injected with sterile PBS, according to body weight.

### 4.2. Sample Collection

Rat lung tissues were collected on the 7th, 14th, 21st and 28th days after MCT injection; samples collected on the same day were assigned into one group and, hence, totally, four groups of samples were obtained. Firstly, the rats were weighed and anesthetized with isoflurane (RWD Life Science, Shenzhen, China). Then, a Millar’s Mikro-Tip catheter transducer (SPR-513) was inserted into the right ventricle through the jugular vein to measure the right ventricular systolic pressure (RVSP). After removing the atria, the lung and heart were perfused with about 20 mL cold PBS to remove the blood. Then, the right middle lung lope was fixed in 4% paraformaldehyde solution for histological analysis and the left lope was snap frozen in liquid nitrogen. For the rat heart, the right ventricular (RV) was dissected from the left ventricular (LV) and the septum (S). The RV and LV with septum were weighed to calculate the Fulton Index, which is equal to RV/(LV + S).

### 4.3. Mass Spectrometry Sample Preparation

The frozen lung tissues were ground in a mortar which was continuously chilled with liquid nitrogen. For each sample, about 50 μL tissue powder was collected and suspended in 400 μL 8.5 M urea buffer (in 50 mM HEPES, pH 8.5), supplemented with protease and phosphatase inhibitor (Thermo, Shanghai, China). The lung homogenates were kept on ice for 30 min and vortexed every 5 min and then centrifuged at 14,000× *g* for 10 min at 4 °C. The supernatant was transferred into a new tube and protein concentration was determined with a BCA Protein Assay kit (TaKaRa, Dalian, China). For proteome analysis, 250 μg total protein was aliquoted and the sample volume was adjusted to 100 μL with lysis buffer; for phosphopeptide enrichment, 1 mg total protein was aliquoted and the sample volume was adjusted to 400 μL. Then, each sample was reduced with 5 mM DTT at 37 °C for 1 h and alkylated with 10 mM iodoacetamide at 25 °C for 30 min. The proteins were precipitated with methanol and chloroform and dried to completeness. Protein precipitation was resuspended either in 100 μL or in 400 μL 1M urea, digested with Lys-C (1:100 enzyme to protein; 125-05061, WaKo, Guangzhou, China) at 25 °C for 2 h and then trypsin (1:50; V5113, Promega, Madison, WI, USA) at 25 °C overnight. After digestion, the peptides mixture was desalted immediately with Oasis HLB 1 cc Vac Cartridge (Waters, Milford, MA, USA).

For peptides labeling, 8 (127N, 127C, 128N, 128C for control samples and 129N, 129C, 130N, 130C for MCT samples) of the 10 isotope channels of a TMT10plex Mass Tag Labeling Reagents kit (90,110, Themo Fisher Scientific, Waltham, MA, USA ) were used. Dried peptides were resuspended either in 25 μL (for proteome analysis) or in 100 μL (for phosphopeptide enrichment) 50 mM HEPES. For each sample, 2 μL peptide solution was used for concentration measurement with the Pierce Quantitative Colorimetric Peptide Assay Kit (23,275, Themo Scientific). For proteome analysis, 80 μg peptides were labeled, and for phosphopeptide enrichment, 400 μg peptides were used. Each sample was labeled with one TMT channel at room temperature for 1 h and then the reaction was quenched with 5% hydroxyammine (Sigma, Shanghai, China). After TMT labeling, all the samples of the same group were mixed and desalted either with the 1cc Oasis HLB column (WAT094225, Waters) or the 3 cc Sep-Pak tC18 Vac Cartridge (WAT054925, Waters).

### 4.4. Peptide Fractionation

TMT-labeled peptide mixture was subjected to offline basic reversed phased fractionation on the UltiMate 3000 HPLC instrument (Thermo Scientific), which was equipped with an automated fraction collector and connected with the XBridge BEH C18 Column (130 Å, 3.5 µm, 4.6 mm × 250 mm; 186,003,943, Waters). Buffer A contained 4.5 mM ammonium formate (pH 10) and 3% acetonitrile; buffer B contained 4.5 mM ammonium formate (pH 10) and 90% acetonitrile. A 72 min gradient was performed and the flow rate was set at 1 mL per minute. Totally, 72 fractions were collected and further pooled in 15 mL tubes to form 12 final fractions, of which the first final fraction was produced by combining original fractions 1, 13, 25, 37, 49, 61. The other fractions were processed using a similar strategy. These final fractions were then evaporated to completeness in a freeze dryer (LyoQuest-85 plus, Telstar, Concord, CA, USA). Each dried peptide fraction was reconstituded in 100 μL 0.1% formic acid (FA) and 1 μL was used for LC–MS/MS analysis. For samples for phosphopeptide enrichment, 3% of each final fraction was put aside as input for proteome analysis. These samples were dried using vacuum-centrifuge and reconstituted in 0.1% FA.

### 4.5. Phosphopeptide Enrichment

The remaining 97% peptide fractions for phosphopeptide enrichment were further combined into 6 fractions. Each of these 6 fractions was firstly enriched with the High-Select™TiO_2_ Phosphopeptide Enrichment Kit (A32993, Thermo Scientific). Flowthroughs from binding and washing steps were combined and further enriched with the High-Select Fe-NTA Phosphopeptide Enrichment Kit (A32992, Thermo Scientific). Elutions containing enriched phosphopeptides were dried completely and reconstituted in 10 μL 0.1% FA before submission to LC–MS/MS analysis.

### 4.6. Mass Spectrometry (MS) Data Acquisition

Reconstituted peptides for proteome analysis and enriched phosphopeptides were loaded into an Easy-nLC-1200 HPLC system connected to an Orbitrap Fusion Lumos Tribrid Mass Spectrometer (Thermo Fisher Scientific). For peptides separation, a 15 cm C18 column (HS-Anal-C-5U-15CM, Beijing Happy Science Scientific, Beijing, China ) was used; the mean particle size was 5 μm and pore diameter was 120 Å. Peptides were separated at a flow rate of 300 nL/min and an 80 min gradient was performed. Buffer A contained 0.1% FA; Buffer B contained 0.1% FA and 90% acetonitrile. For proteome analysis, an MS2 mode quantification method was used. MS1 spectra were collected with an Orbitrap detector using these parameters: resolution 120,000, scan range 350–1800 (*m*/*z*), AGC target 4 × 10^5^, 50 ms maximum injection time (MIT), included charge states 2–6. MS2 spectra were collected in data-dependent mode, in which the top 12 precursors were scanned, and the parameters included quadrupole isolation mode, 38% high energy collision dissociation (HCD) energy, auto scan range mode, 50,000 orbitrap resolution, 1 × 10^5^ AGC target. For phosphopeptides analysis, Synchronous Precursor Selection (SPS)–MS3 method was used. MS1 spectra were collected with these parameters: Orbitrap detector, 120,000 resolution, 375–1500 (*m*/*z*) scan range, 50 ms MIT, 4 × 10^5^ AGC target. MS2 spectra were collected in an ion trap (IT) with these parameters: quadrupole isolation mode, collision-induced dissociation (CID) activation, 35% CID energy, Turbo IT scan mode, 1 × 10^4^ AGC target, 50 ms MIT, 400–1200 (*m*/*z*) scan range. The top 10 SPS precursors were submitted to MS3 scan with these parameters: HCD activation, 65% HCD energy, Orbitrap detector, 60,000 resolution, 100–800 (*m*/*z*) scan range, 120 ms MIT, 1 × 10^5^ AGC target.

### 4.7. MS Data Analysis 

MS raw files were analyzed using the Proteome Discoverer software 2.2 (Thermo Fisher Scientific). A template processing workflow (PWF_Fusion_Reporter_Based_ Quan_MS2_SequestHT_Percolator) within the software was applied. Data were searched against the rat reference proteome database (UP000002494, 29,934 proteins), downloaded from UniProtkb, and the contaminant database. To avoid interference by the isotope impurities, isotopic correction factors were applied according the lot number of the TMT reagent. Trypsin (full) was chosen as the digestion enzyme, maximal missed cleavage was set at 3 and minimal peptide length was 6. Precursor mass tolerance was set at 10 ppm and fragment mass tolerance was 6 Da. Static modifications included cysteine residue carbamidomethylation (+57.021 Da), N-terminus and lysine residue TMT-6plex (+229.163 Da). Dynamic modifications included N-terminus acetylation (+42.011 Da) and methionine residues oxidation (+15.995 Da). For phosphopeptides analysis, a PWF_Fusion_TMT_Quan_SPS_MS3_SequestHT_Percolator workflow was used. Most of the search parameters remained the same, except that additional dynamic phospho/+79.966 Da modifications on serine, threonine and tyrosine residues were added.

For proteome analysis, the sum of all the Scaled Abundance values in each channel was used to normalize protein loading across different channels. Hence, for specific protein, its normalized Scaled Abundance was regarded as its expression level and used for quantification and statistical analysis. Phosphopeptide quantification and analysis were performed following the same method.

### 4.8. Statistical and Bioinformatics Analysis

Statistical and bioinformatics analysis were mainly performed in R Studio, GraphPad Prism or Excel. Student’s *t*-test was used to compare the quantitative data between two groups. Data were presented as mean ± standard deviation (SD). For volcano plots, data were processed in Perseus 1.6.15.0 and the exported matrices were visualized with ggplot2 package. Gene Ontology analysis was performed with clusterProfiler package [41] and Kyoto Encyclopedia of Genes and Genomes (KEGG) pathway analyses were performed with Metascape [42]. Protein–protein interaction networks were analyzed online in String and visualized in Cytoscape 3.8.2 [43].

## Figures and Tables

**Figure 1 ijms-24-09629-f001:**
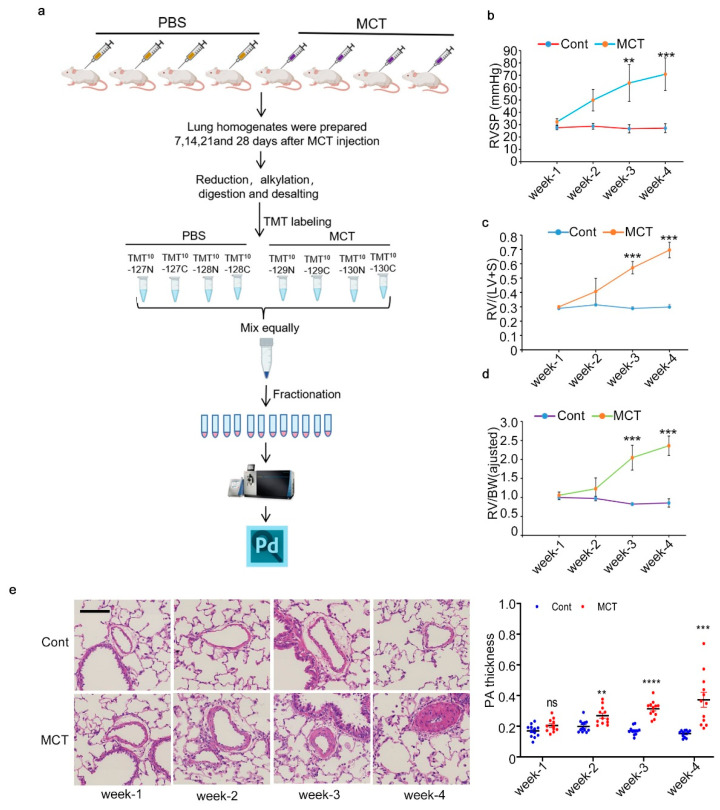
Study design and hemodynamic analysis of MCT-induced PAH rats. (**a**) Experimental design and workflow for TMT-based quantitative proteomic analysis of lung tissues from rats treated with MCT. (**b**) Right ventricular systolic pressure (RVSP) of rats treated with MCT or vehicle control (Cont) for the indicated weeks. (**c**) Changes in Fulton index, which is equal to right ventricle/(left ventricle + septum). (**d**) Line chart showing the changes in RV/body weight. (**e**) Representative HE staining images of lung sections from rats treated with MCT or vehicle control and the statistical results of pulmonary arterial wall thickness. Scale bar = 100 μm. For (**a**–**e**), results represent mean ± SD, ns means not significant, ** *p* < 0.01, *** *p* < 0.001, **** *p* < 0.0001, unpaired Student’s *t*-test.

**Figure 2 ijms-24-09629-f002:**
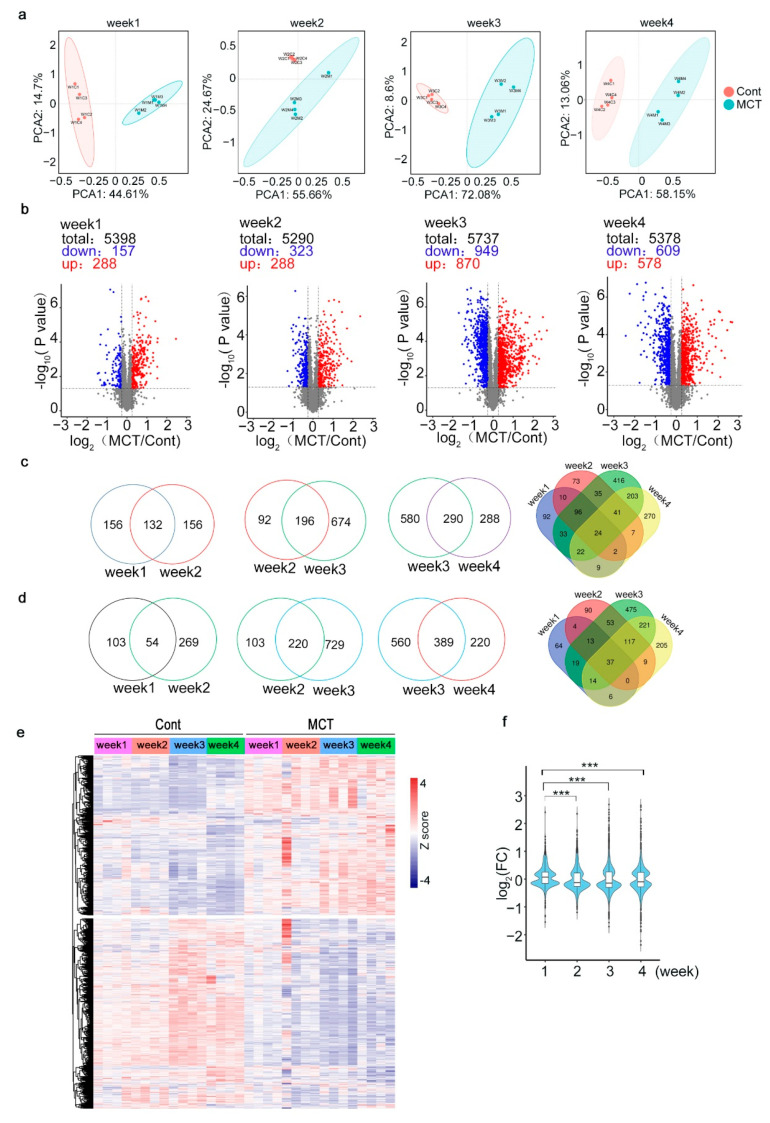
Proteomic profiling of the MCT-treated rats lung tissues. (**a**) Principal component analysis (PCA) of scaled protein abundances for all biological replicates of control (Cont) and MCT-treated samples. (**b**) Volcano plots showing the changes in proteins in Cont and MCT samples of each group. (**c**,**d**) Venn diagrams showing the numbers of proteins significantly upregulated (**c**) or downregulated (**d**) in MCT samples compared to control samples. (**e**) Heatmap showing the protein abundances of all samples. Proteins changed (*p*-value < 0.05, fold change < 0.83 or >1.2) in at least one group of data were included. (**f**) Violin plots showing the log_2_-transformed fold changes of protein abundance of MCT to Cont for each group. *** *p* < 0.001.

**Figure 3 ijms-24-09629-f003:**
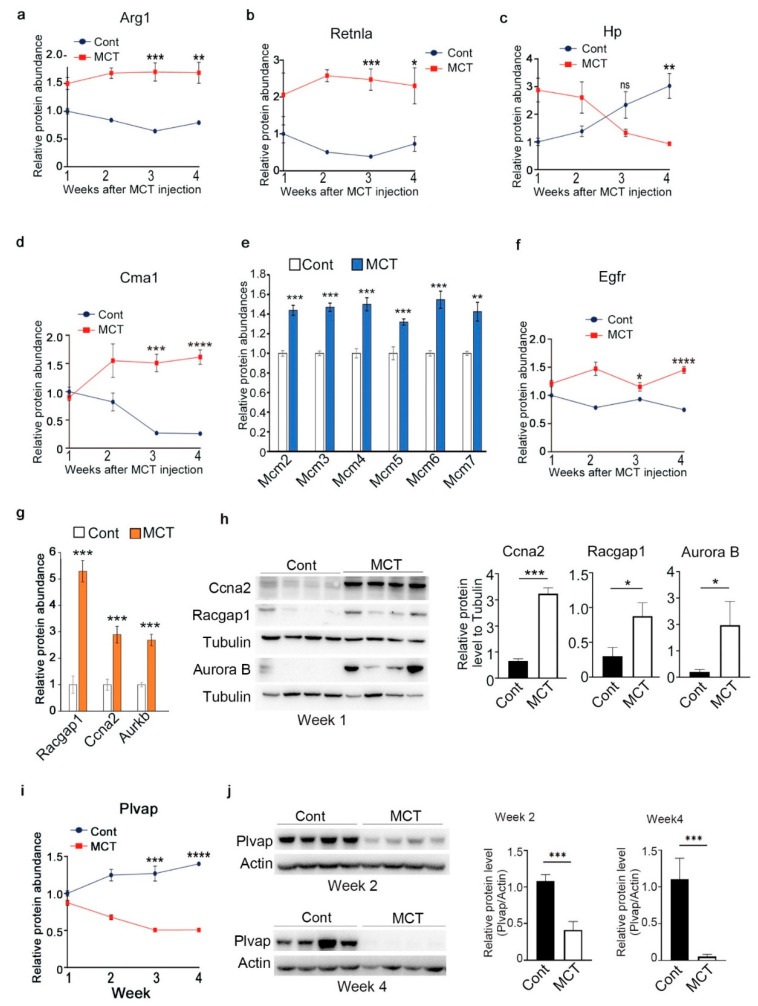
Verification of representative proteins identified in proteomic analysis of the lung tissue. (**a**–**d**,**f**,**i**) Relative abundance of the indicated proteins identified by proteomic analysis in lung tissues from rats treated with MCT or PBS (Cont). (**e**,**g**) Relative abundances of the indicated proteins upregulated in samples from rats treated with MCT for one week. (**h**,**j**) Western blot (WB) analysis and quantification of the indicated proteins in lung tissues of rats treated with MCT or PBS. Data represent mean ± SD; ns means not significant, * *p* < 0.05, ** *p* < 0.01, *** *p* < 0.001, **** *p* < 0.0001, n = 4.

**Figure 4 ijms-24-09629-f004:**
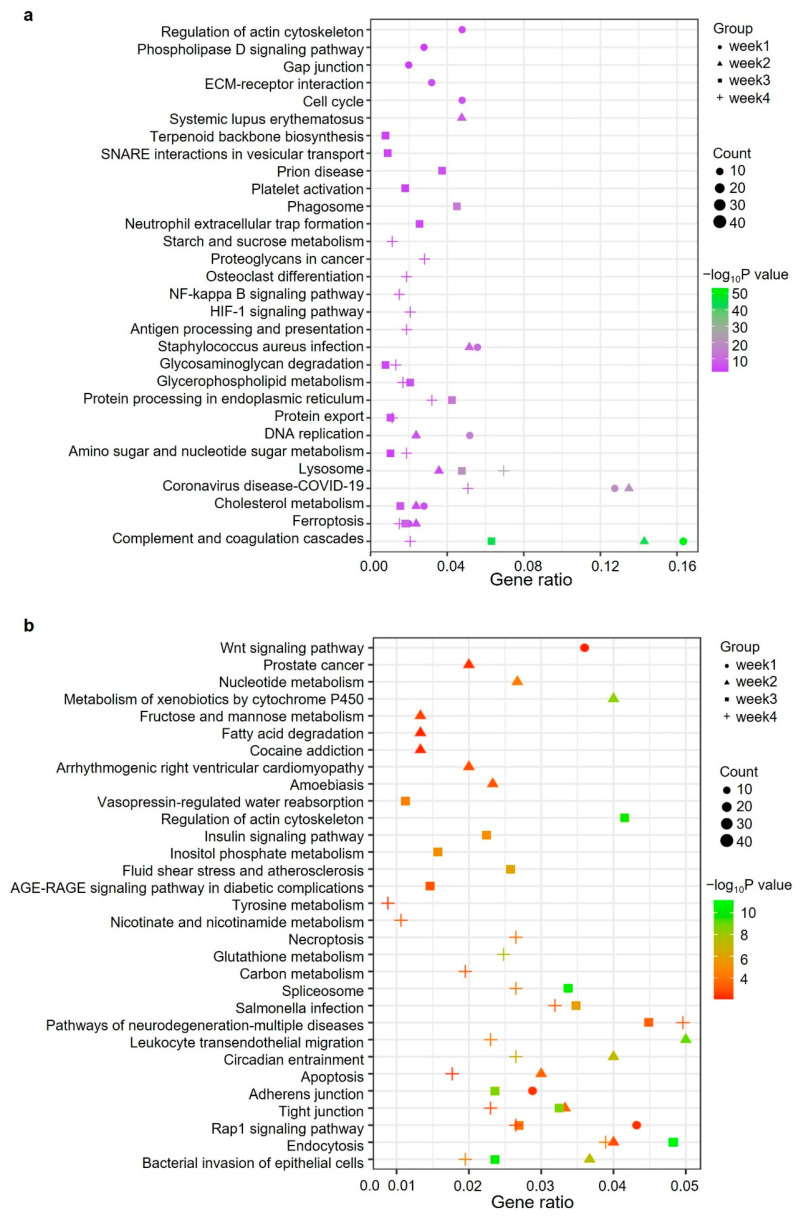
Pathway analysis of differently expressed proteins. KEGG pathway analysis of proteins significantly upregulated (FC > 1.2, *p*-value < 0.05) (**a**) or downregulated (FC < 0.83, *p*-value < 0.05) (**b**) in lung tissues of MCT-treated rats compared to control rats.

**Figure 5 ijms-24-09629-f005:**
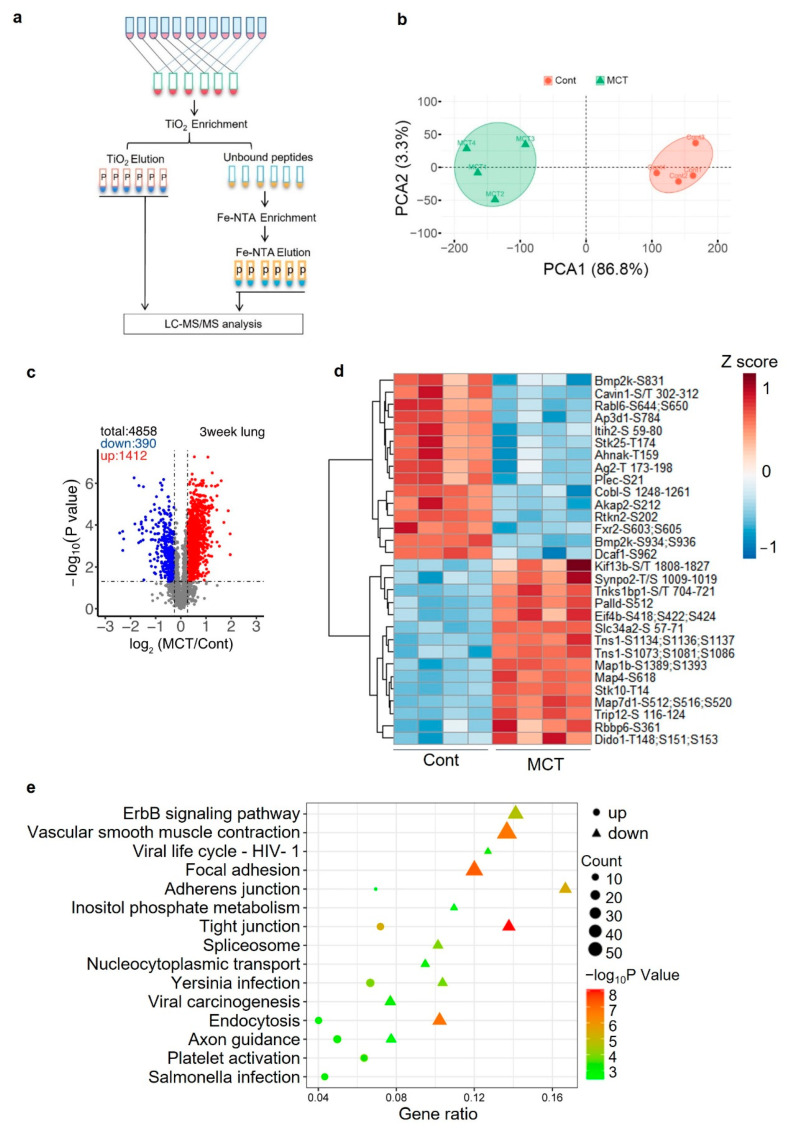
Phosphoproteomic profiling of lung tissues from MCT-induced PAH model rats. (**a**) Workflow for phosphopeptides enrichment for lung tissues. (**b**) PCA analysis for phosphopeptides’ abundances for all biological replicates of control (Cont) and MCT treated samples. (**c**) Volcano plots showing the changes in phosphopeptides in lung tissue of Cont and MCT rats. (**d**) Heatmap showing the top-15 downregulated phosphopeptides (FC < 0.83, *p*-value < 0.05) and upregulated phosphopeptides (FC > 1.2, *p*-value < 0.05). (**e**) Pathway enrichment analysis for significantly changed phosphoproteins in lung tissues from rats treated with MCT or vehicle control for three weeks.

**Figure 6 ijms-24-09629-f006:**
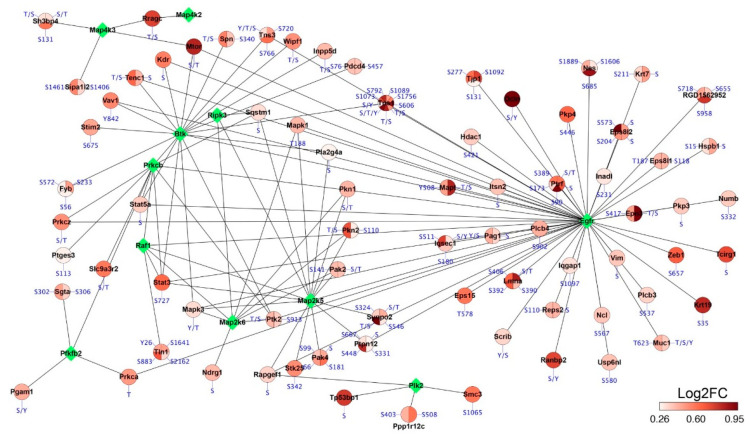
PPI network between changed kinase proteins and changed phosphoproteins. PPI networks were created for significantly upregulated kinase proteins and upregulated phosphoproteins identified in the lung tissue from rats treated with MCT for three weeks. Nodes in green indicate kinase proteins identified in proteome analysis.

## Data Availability

The mass spectrometry proteomics data have been deposited to the ProteomeXchange Consortium (http://proteomecentral.proteomexchange.org) via the iProX partner repository [44] with the dataset identifier PXD038462 (accessed on 30 November 2022).

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
