# Peer review of "Quantitative Proteomic and Phosphoproteomic Profiling of Lung Tissues from Pulmonary Arterial Hypertension Rat Model"

_ijms, 2023, doi:10.3390/ijms24119629_

Round 1

Reviewer 1 Report

Comment to the authors ijms-2382325.

In this study Ang Luo and colleagues aimed to identify the proteins and pathways involved in the progression of pulmonary arterial hypertension (PAH) using monocrotaline-treated rats as a model. The study found 2660 significantly changed proteins, including known PAH-related proteins, and verified the expression of potential PAH-related proteins using western blot. Phosphoproteomic analysis also identified upregulated and downregulated phosphopeptides potentially involved in the disease progression/etiology. Strengths of the study include the comprehensive analysis of proteins and phosphoproteins, while weaknesses include the use of a single animal model and the limited application of findings to humans. It is a well-written study, however, several main concerns should be addressed. 

1)    In the abstract, it would be helpful to include the absolute fold change threshold used to identify significantly changed proteins. 

a.     Similarly, in the text It is not clear whether the authors used a p-value <0.05 or an adjusted p-value <0.05. The authors should clarify this point and justify their choice of cutoff.

b.     It is not clear whether the cutoff for protein-protein interaction network analysis was log2(FC)|> log21.5 or adj p-value <0.05 and log2(FC)|> log21.5. The authors should clarify this point.

2)    While Retnla and arginase-1 are known to be PAH-related proteins, it is concerning that the authors did not observe changes in expression of key players such as STAT3, HIF1a, and IL-6. It would be helpful for the authors to provide a comment on this point.

3)    The study focuses on changes in protein expression in the lungs, but it may be interesting to also compare changes in the pulmonary arteries. This point should be discussed. 

4)    It would be useful for the authors to compare their findings to publicly available RNA seq databases for monocrotaline lungs, such as GSE149713, and to compare their findings to proteomic data from hypoxia-induced PAH rat models. Similarly the authors should compare their finding to what was previously published in the field (e.g. PMID: 36388115)

5)    Regarding the animal model :

a.     The authors used a dose of 50mg/kg to induce PAH, which is lower than the typical dose of 60mg/kg. It would be helpful for the authors to provide a justification for their choice of dose.

b.     It would be useful for the authors to provide a more comprehensive characterization of the animals, including cardiac output and total pulmonary resistance. This weakness should be acknowledged in the limitations section.

c.     There is variability in MCT-PAH severity at 4 weeks, and the authors selected only 4 rats at each time point for proteomic analysis. It would be helpful to know the disease severity in these animals.

6)    The authors mention that "6901 master proteins were identified and 6759 proteins were quantified, and 4263 quantified proteins were shared among 4 groups of data." It is unclear what is meant by "master proteins." The authors should clarify this point.

7)    It would be helpful for the authors to provide a principal component analysis (PCA) that includes all groups (control, MCT week 1, 2, 3, and 4) to assess whether the MCT proteome changes over time and with disease severity.

8)    To better understand proteins associated with disease progression/severity, it may be interesting to perform a correlation matrix between protein expression, hemodynamics, and vascular remodeling.

9)    Regarding the gene ontology analysis presented in Figure 4.

a.     The adjusted p-value used for gene ontology analysis is quite low. 

b.     Additionally, it is concerning that the pathways involved at 1, 2, 3, and 4 weeks are all different, inflammation seems to be decreased in MCT lungs (contrary to expectations), oxidative phosphorylation seems to be increased (which does not fit with the Warburg effect observed in PAH), and no gene ontology supports the potential "cancer-like" theory observed in PAH. 

c.     The authors should carefully analyze and discuss their gene ontology data, and consider these findings with caution.

10) It would be helpful to perform a PCA analysis for the phosphoproteomics data, and to investigate any correlations or associations between phosphorylated protein and protein levels. Additionally, it is concerning that critical players known to be involved in PAH, such as phospho-Stat3, did not show up in the analysis. The authors should comment on this point.

11) The paragraph on pages 13-14 is based on Figure 6. There is no figure 6 available in the manuscript. 

Reviewer 2 Report

The present study analyzed the proteins and phosphoproteins involved in pulmonary arterial hypertension (PAH) in a comprehensive way, using the lung tissues from rats with monocrotaline-induced PAH. Overall, this is well designed study, methods are modern and appropriate, and a notable amount of novel and significant results has been obtained. However, there are several major concerns that need to be clarified and modified in order to be suitable for publication.

1. Detailed language check needs to be performed. There are too many errors, and some sentences are difficult to understand. Some of the examples are:

Line 8 – it should be ‘fatal’ instead of ‘fetal’!

Line 10 – arteries instead of arterials.

Title – why Pulmonary with capital P?

The use of singular and plural is not correct (attention instead of attentions, pathways instead of pathway, death among patients instead of patient death, etc).

Line 52 – TMT should be defined here since it is the first time mentioned.

There are many more similar examples.

2. Abstract needs to be modified in order to be more comprehensible. For example, the part ‘(abslog2(fold change)14 > log2[1.2], P <0.05)’ is redundant. The last sentence cannot be a conclusion, please modify that.

3. Specify statistical tests in Methods.

4. Why did you use standard deviation (SD) for some results (as in Fig 1) and standard error of the mean (SEM) for some other results (as in Fig 3)? It should be uniform throughout the text.

5. You should elaborate how you chose the treatment period for phosphoproteomics profiling of lung tissue. Why did you perform phosphophoproteomic profiling with the lung tissue from rats treated with MCT for 3 weeks, and not 1, 2 or 4 weeks?

6. There is no Figure 6 in the text (that the authors are referring to)!

Detailed language check needs to be performed.

Round 2

Reviewer 1 Report

Overall, the authors have made significant improvements in addressing my comments and acknowledging the study's limitations in the appropriate section.

However, I still have concerns regarding the use of a cutoff value of P<0.05. A more conservative approach would be to utilize an adjusted p-value < 0.05. It is important to acknowledge this in the limitation section to ensure transparency.

Furthermore, I am troubled by the lack of overlap between the previously published RNA sequencing data and the findings from the proteomic/phosphoproteomic analysis. Although the authors have acknowledged this discrepancy, the absence of key players in PAH within their proteomics data remains puzzling.

Author Response

Overall, the authors have made significant improvements in addressing my comments and acknowledging the study's limitations in the appropriate section. 

However, I still have concerns regarding the use of a cutoff value of P<0.05. A more conservative approach would be to utilize an adjusted p-value < 0.05. It is important to acknowledge this in the limitation section to ensure transparency. 

Response: Thank you very much for your comments. In the revised manuscript we added “To identify the significantly changed proteins or phosphopeptides, we used a cutoff value of p < 0.05 instead of an adjusted p-value < 0.05. The benefit of doing so is that more proteins or phosphopeptides could be identified, but it also increased the risk of discovering more false positive proteins or phosphopeptides. Therefore, extra caution should be paid to these significantly changed proteins or phosphopeptides with relatively small FC or big p-value when doing further verification studies.” (Lines 469-474)

Furthermore, I am troubled by the lack of overlap between the previously published RNA sequencing data and the findings from the proteomic/phosphoproteomic analysis. Although the authors have acknowledged this discrepancy, the absence of key players in PAH within their proteomics data remains puzzling.

Response: We apologize for having not clearly described the comparison between our results and previously published studies. From line 249 to line 261, we mentioned “It’s a little strange that no overlap between these two studies for rats treated with MCT for one week. But from Week 2 to Week 4, the number of proteins shared by these two studies kept increasing. Representative common upregulated genes or proteins included Arg1(arginase 1), chymase 1 (Cma1), fibronectin 1 (Fn1) and platelet factor 4 (Pf4); representative common downregulated genes or proteins included vasoactive intestinal peptide receptor 1(Vipr1), angiotensin I converting enzyme (Ace I) and calcitonin receptor-like receptor (Calcr1) (Supplementary Table S1). This discrepancy between our study and Xiao et al.’s study may partly come from the difference in experimental design. In our study, each group of MCT-treated rats had the corresponding PBS-treated rats as the control sample and a total of four groups of control samples were collected. However, in Xiao et al.’s study, only one group of control rats was included for analysis.” In addition, we did find phosphorylated Stat3-Ser727 as an upregulated phosphoprotein in the MCT-treated rat lung samples. In the revised manuscript we added, “In addition, one of the known PAH-related phosphoproteins Stat3 (Ser-727) was also found to be upregulated (-log10P-value= 5.89, FC=1.5 or log2FC= 0.6) in our data (Supplementary Table 3), indicating consistency between our findings and previous publications.”. (Lines 451-454)

Reviewer 2 Report

The authors improved the initial version of the manuscript according to the suggestions raised.

Language is improved; however, there are still several grammatical, linguistic and other language-related errors that should be corrected.

Author Response

Comments and Suggestions for Authors
The authors improved the initial version of the manuscript according to the suggestions raised.

Response: Thank you for your recognition of our revised manuscript.

Comments on the Quality of English LanguageLanguage is improved; however, there are still several grammatical, linguistic and other language-related errors that should be corrected.

Response: Thank you for your reminder. In the current revised manuscript, we go through the manuscript carefully and corrected the grammatical, linguistic, and other language-related errors as far as we can.